**Data Availability Statement:** All relevant data are within the paper and its Supporting Information files.

# Clinical outcomes of immunoglobulin treatment for patients with secondary antibody deficiency: Data from the Ontario immunoglobulin treatment case registry

Armin Abadeh[1]☯, Sarah Shehadeh[2]☯, Stephen Betschel[1], Susan Waserman[3], Donald William Cameron🔢[2,4], Juthaporn Cowan🔢[2,4]*

1 Department of Medicine, University of Toronto, Toronto, Ontario, Canada, 2 Clinical Epidemiology Program, The Ottawa Hospital Research Institute, Ottawa, Ontario, Canada, 3 Department of Medicine, McMaster University, Hamilton, Ontario, Canada, 4 Department of Medicine, University of Ottawa at The Ottawa Hospital, Ottawa, Ontario, Canada

☯ These authors contributed equally to this work.
* jcowan@toh.ca

## Abstract

### Background

Despite the increasing number of cases of secondary antibody deficiency (SAD) and immunoglobulin (Ig) utilization, there is a paucity of data in the literature on clinical and patient-reported outcomes in this population.

### Objective

To describe immunoglobulin utilization patterns, clinical and patient-reported outcomes in patients with SAD on immunoglobulin replacement therapy (IgRT).

### Methods

A cross-sectional study of patients with secondary antibody deficiency enrolled in the Ontario Immunoglobulin Treatment (ONIT) Case Registry from June 2020 to September 2022 was completed. Demographics, comorbidities, indications for immunoglobulin treatment, clinical infections at baseline and post IgRT, and patient-reported outcomes were collected and analyzed.

### Results

There were 140 patients (58 males; 82 females; median age 68) with SAD during the study period; 131 were on subcutaneous Ig (SCIG) and 9 were on intravenous Ig (IVIG). The most common indication was chronic lymphocytic leukemia (CLL) (N = 52). IgRT reduced the average annual number of infections by 82.6%, emergency room (ER) visits by 84.6%, and hospitalizations by 83.3%. Overall, 84.6% of patients reported their health as better compared to before IgRT. Among those patients who switched from IVIG to SCIG (N = 35), 33.3% reported their health as the same, and 62.9% reported their health as better.

**Funding:** JC received honoraria and consultation fees from AstraZeneca, Merck, GSK, Sanofi, Genzyme, Takeda, CSL Behring, Octapharma, and Biogen. SB received advisory, speaker and committee fees and operational research funds to the institution, not in the form of salaries personally from Astria, Canadian Blood Services, CSL Behring, Grifols, Ionis Pharmaceuticals, Kalvista Novartis, Octapharma, Pharvaris, Sanofi, and Takeda. SW received speaker and consultation fees from CAAIF, ALK Abello, Pfizer, Aimmune Schroeder Foundation, Sean Delaney foundation, GSK, Novartis, CSL Behring, Sanofi, Astrazaneca, Takeda, Teva, Medexus, Mylan, AbbVie, Miravohealth, Bausch Lomb, Avir Pharma and Covis. DWC received consultation and speaker fees from Takeda and CSL Behring. SS and AA have declared that no competing interests exist. The funder did not have any additional role in the study design, data collection and analysis, decision to publish, or preparation of the manuscript. This commercial affiliation does not alter our adherence to PLOS ONE policies on sharing data and materials.

**Competing interests:** I have read the journal's policy and the authors of this manuscript have the following competing interests: JC received honoraria and consultation fees from AstraZeneca, Merck, GSK, Sanofi, Genzyme, Takeda, CSL Behring, Octapharma, and Biogen. SB received advisory, speaker and committee fees or research funding from Astria, Canadian Blood Services, CSL Behring, Grifols, Ionis Pharmaceuticals, Kalvista Novartis, Octapharma, Pharvaris, Sanofi, and Takeda. SW received, speaker and consultation fees from CAAIF, ALK Abello, Pfizer, Aimmune Schroeder Foundation, Sean Delaney foundation, GSK, Novartis, CSL Behring, Sanofi, Astrazaneca, Takeda, Teva, Medexus, Mylan, AbbVie, Miravohealth, Bausch Lomb, Avir Pharma and Covis. DWC received consultation and speaker fees from Takeda and CSL Behring. SS and AA have declared that no competing interests exist.

## Conclusions

This study demonstrates that IgRT significantly improved clinical outcomes and patient-reported general health state in patients with SAD. This study also further supports the use of SCIG in patients with SAD.

## Introduction

Secondary antibody deficiency (SAD) is distinct from primary immunodeficiency disorder (PID), which is also known as inborn errors of immunity (IEI) and encompasses more than 400 inherited disorders of immunity. SAD is caused by a quantitative or qualitative decrease in antibodies that can result from renal or gastrointestinal Ig loss, hematological malignancies such as chronic lymphocytic leukemia (CLL), lymphoma and multiple myeloma (MM), and immunosuppressive medications such as steroids, rituximab and other B-cell depleting agents [1–3]. Compared to PID, SAD prevalence is estimated to be 30-times higher, but SAD may be reversible if the underlying cause is treated [4]. Previous studies have reported hypogammaglobulinemia in approximately 25% of patients with newly diagnosed CLL and significant humoral dysfunction during CLL in additional 25% of patients with normal Ig levels at the time of diagnosis [5]. Similarly, immunomodulatory and immunosuppressive therapies either in combination with other treatments or as maintenance therapies have been associated with increasing prevalence of SAD. For example, hypogammaglobulinemia has been identified in 38.5% of patients with initially normal Ig levels following treatment with rituximab [6].

Since its first practical use in 1950s, IgRT has become a well-established treatment for patients with PID, significantly reducing the incidence of infections such as pneumonia, bronchitis, sinusitis, ear infections, meningitis and skin infections [7–9]. However, even in the current guidelines, there is a paucity of evidence-based recommendations related to IgRT in patients with SAD. Previous SAD clinical trials were conducted prior to the introduction of more advanced immunosuppressive and immunomodulatory therapies [10, 11]. In addition, while data for subcutaneous immunoglobulin (SCIG) in PID suggests that SCIG is associated with fewer systemic side effects, more stable Ig trough levels and improvement in quality-of-life parameters compared to IVIG, there is a lack of guideline-specific recommendations for the use of SCIG in patients with SAD [10, 12, 13].

The wider and more effective use of immunosuppressive medications in the treatment of malignancies and autoimmune conditions is resulting in increased survival among this patient population, and an increase in the prevalence of SAD. It is therefore becoming increasingly important to address this unmet need in this growing patient population by optimizing IgRT in these patients.

Ontario Immunoglobulin Treatment (ONIT) program, is a multicentre program, based at three teaching hospitals across Ontario, with ongoing funding by the Ministry of Health. The program mandates improving healthcare delivery as well as training and supporting home-based SCIG. As part of this program, a clinical case registry has been created as a tracking and performance-based evaluation tool to monitor IgRT indication, efficacy, and patient reported outcomes. This is the first report from the ONIT registry and the largest study on clinical and patient-reported outcomes in Canadian patients with SAD receiving IgRT.

## Methods

### Study design and participants

The ONIT Case Registry is a database capturing patient demographics, Ig treatment indication, dosage, regimen, clinical outcomes such as infection, ER visits and hospitalizations, as well as patient-reported outcomes that are prospectively collected at follow-up visits with the physician every 6–12 months. For this study, we included data of all adult patients (age $\geq$ 18 years old) enrolled in the ONIT case registry between June 1$^{st}$ 2020 and September 30$^{th}$ 2022, with diagnoses of antibody deficiency secondary to CLL, lymphoma, MM and other plasma cell dyscrasias, solid organ transplant (SOT), hematopoietic stem cell transplant (HSCT), and patients on immunosuppressive therapy for autoimmune conditions such as rheumatoid arthritis and Crohn's disease. Patients receiving Ig for treatment of immune-mediated neurological conditions such as chronic inflammatory demyelinating polyneuropathy and myasthenia gravis were excluded from this study as Ig administration in neurological conditions is used for immunomodulation rather than as replacement therapy. Data used in this study was extracted on September 30$^{th}$, 2022. There were 7 SAD patients (5.0%) who passed away before this time. The data closest to their time of death was extracted for analysis.

Patients were asked to fill out a baseline visit questionnaire at the start of the study about infection rate and hospitalizations as well as general health state prior to receiving any Ig treatment. At each follow-up visit, patients filled out questionnaires with the clinical team reporting on infections, ER visits, hospitalizations, as well as their state of health after being on IgRT. Patients were asked if their overall health was the same, worse, or better compared to before IgRT. Additionally, similar questions were asked of patients who switched their IgRT from IV to SC or from SC to IV. Questionnaires were completed using a tablet, computer, or pen and paper. Patients were also asked to fill out personal diaries at home, keeping a log of their weekly infusions where they recorded the LOT number of the product used, the sites of infusion and any adverse reactions they developed.

A centralized ethics approval was first obtained at the provincial level from Clinical Trials Ontario (CTO #1978). Institutional approval was then obtained at each study site. Informed written or verbal consent was obtained from each patient participating in the study. A note-to file was created to document verbal consents.

### Outcomes

The outcomes of interest were immunoglobulin therapy dosage, annual number of infections, ER visits, hospitalizations, as well as patient-reported outcomes.

### Data analysis

Descriptive data analysis was performed using Microsoft Excel 2020. Wilcoxon signed rank test was used to compare continuous variables before and after IgRT initiation. Spearman's correlation was used to measure the strength and direction of relationship between baseline neutrophil counts and infection rate.

## Results

### Demographics

Table 1 summarizes the demographics and baseline data. A total of 140 patients (58 males; 82 females; median age 68) with SAD were identified. Of these patients, 131 were on SCIG (of whom 35 were previously on IVIG) and 9 were on IVIG (3 of whom were previously on

**Table 1. Demographic profile of study participants.**

| Demographic | Result |
|---|---|
| Male (N, %) | 58 (41.4%) |
| Female (N, %) | 82 (58.6%) |
| Age (years) Median [Q1, Q3] | 68 [61, 73] |
| Age range | 23–85 years |
| IVIG (N, %) | 9 (6.4%) |
| Previously on SCIG (N, %) | 3 (33%) |
| SCIG (N, %) | 131 (93.6%) |
| Previously on IVIG (N, %) | 35 (26.7%) |
| Followed by other subspecialists (%) | |
| Hematology Oncology (%) | 63.4% |
| Respirology (%) | 11.9% |
| Other: Cardiology, Rheumatology, Endocrinology, Nephrology (%) | 27.5% |
| Primary Disease: | |
| Chronic Lymphocytic Leukemia (N, %) | 52 (37.1%) |
| Lymphoma (N, %) | 33 (23.6%) |
| Multiple Myeloma / Monoclonal | 20 (14.3%) |
| gammopathies (N, %) | |
| Waldenstrom macroglobulinemia (N, %) | 6 (4.3%) |
| Transplantation (N, %) | 12 (8.6%) |
| Hematopoietic Cell Transplant | 1 |
| Kidney transplant | 5 |
| Lung transplant | 4 |
| Liver transplant | 1 |
| Heart transplant | 1 |
| Immunosuppressive therapy for | 17 (12.1%) |
| autoimmune diseases: | |
| Rheumatoid Arthritis (N, %) | 7 (41.2%) |
| Median [Q1, Q3] interval from transplant to IgRT (years) | 3.0 [2.0, 4.8] years |
| IgRT Dosage (N = 140) | |
| IVIG (g/kg/4 weeks) Median [Q1, Q3] | 0.48 [0.44, 0.51] |
| IVIG (g/4weeks) Median [Q1, Q3] | 40.0 [35.0, 40.0] |
| SCIG (g/kg/4 weeks) Median [Q1, Q3] | 0.44 [0.39, 0.53] |
| SCIG (g/4weeks) Median [Q1, Q3] | 32.0 [24.0, 40.0] |

SCIG). The most common underlying diseases were CLL (N = 52, 37.1%), followed by lymphoma (N = 33, 23.6%%) and plasma cell dyscrasias including Waldenstrom's macroglobulinemia, multiple myeloma, and monoclonal gammopathy of undetermined significance (N = 26, 18.6%).

Among 26 (18.6%) patients with monoclonal gammopathies, the most common type was IgM gammopathy (N = 11, 42.3%), followed by IgG gammopathy (N = 9, 34.6%) and IgA gammopathy (N = 6, 23.1%).

There were 11 (7.9%) SOT and 1 (0.7%) HSCT patients. The most common SOT was kidney transplant (KT) (N = 5, 45.4%), 2 of whom had prolonged immunosuppressive therapies (>10years) for post-transplant lymphoproliferative disorder and chronic organ rejection. The median interval between transplant and onset of IgRT was 3.0 [2.0, 4.8] years for all patients (N = 12) and 3.0 [2.0, 3.0] years for patients after transplant without prolonged immunosuppression (N = 10).

The majority of patients (75.0%) were followed by one or more subspecialists, the most common being hematology/oncology (81.9%).

12 (8.6%) patients had other concurrent medical problems with 2 (16.7%) patients having bronchiectasis, and 10 (83.3%) patients having renal impairment.

## Prior use of immunomodulatory or immunosuppressive agents

The most common therapies received prior to starting IgRT were chemotherapy and B-cell depleting agents; 20.0%% of patients received chemotherapy within 12 months of starting IgRT, 33.6% the year prior; and 20.7%% of patients on rituximab or other within 12 months of starting IgRT and 27.2%% in the year prior. Immunosuppressants such as tacrolimus and mycophenolate were administered to 11.4%% of patients, and systemic corticosteroids such as prednisone $\geq$ 20 mg/day for more than 2 weeks were administered to 9.3% of patients within 12 months of starting IgRT.

## IgRT dosage, formulations, and treatment duration

The median dose of SCIG was 0.44 [0.39, 0.53] g/kg/4 weeks and that of IVIG was 0.48 [0.44, 0.51] g/kg/4weeks. The median duration of IgRT was 3.0 [2.0, 6.0] years.

122 (93.1%) patients were using SCIG 20% formulation with 71 (58.2%) on Cuvitru and 51 on Hizentra (41.8%), and 9 (6.9%) patients were using 16.5% Cutaquig. Facilitated SCIG was not yet available for patients in Canada.

## Serum immunoglobulin levels

The median level of IgG in all patients with SAD was 3.70 [2.35, 4.85]g/L before IgRT and 8.00 [6.45, 9.60] g/L after IgRT initiation (Table 2). The median interval between baseline and most recent IgG measurement was 2.0 [1.0, 5.0] years.

## Clinical outcomes

Of 140 patients, 97 (69.3%) had at least one follow-up visit by September 30, 2022. Sixty-eight patients (70.1%) reported having no infections while 29 patients (29.9%) reported at least one infection. Of 29, 26 (89.6%) were actively receiving SCIG, 1 (3.5%) patient was on IVIG, while 2 (6.9%) patients had discontinued IgRT. There were 87 infections reported. The most common infection was COVID-19 (N = 17, 19.3%), followed by sinusitis and other respiratory tract infections and bronchitis (N = 13, 14.7%). Of note, none of the COVID cases required hospital admission. Other common infections included urinary tract infections (N = 10, 11.4%) and skin and soft tissue infections (N = 9, 10.2%). Pneumonia (N = 2, 2.3%) was reported in two patients who at the time, had discontinued IgRT. After IgRT, patients did not experience severe bacterial infections such as sepsis, meningitides, or osteomyelitis.

Overall, IgRT treatment reduced average annual number of infections by 82.6% (4.6 vs 0.8), average number of ER visits by 84.6% (1.3 vs 0.2), and average number of hospitalizations by 83.3% (0.6 vs 0.1), (Fig 1A). For a subset of patients who were initially started and remained on

**Table 2. Serum Ig levels.**

| | Before IgRT | After IgRT | *p*-value |
|---|---|---|---|
| IgG (n = 107) Median [Q1, Q3] | 3.70 [2.35, 4.85] g/L | 8.00 [6.45, 9.60] g/L | <0.0001 |
| IgA (n = 98) Median [Q1, Q3] | 0.30 [0.10, 0.60] g/L | 0.30 [0.10, 0.53] g/L | 0.626 |
| IgM (n = 99) Median [Q1, Q3] | 0.20 [0.10, 0.50] g/L | 0.20 [0.10, 0.50] g/L | 0.176 |

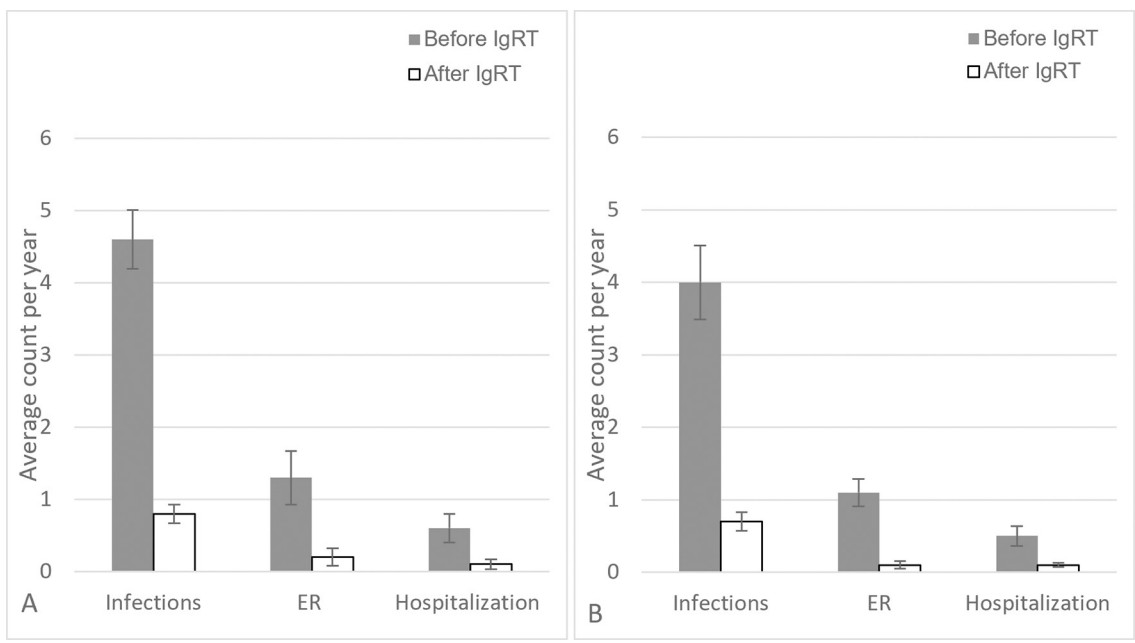

**Fig 1. Clinical outcomes reported before and after IgRT.** Average clinical outcomes (±standard error of mean) reported per year before and after IgRT initiation. (A) in all patients on IVIG or SCIG. (B) in patient subset who has only been on SCIG.

SCIG (N = 95), 47.4% (N = 45) reported an 82.5% decrease in infection rate (4.0 vs 0.7), a 90.9% decrease in ER Visits (1.1 vs 0.1) and an 80.0% decrease in hospitalizations (0.5 vs 0.1) after starting SCIG. (Fig 1B). Of 140 patients, data on baseline neutrophil counts was available in 83 patients. Neutrophils count prior to IgRT were not correlated with baseline infection rate, spearman r = -0.075 [95% CI -0.292–0.1494] (Fig 2).

## Patient-reported outcomes

Overall, 55.7% of patients reported their health state after IgRT, with 84.6% of patients reporting their health as better (Fig 3A). Among those 35 patients who switched from IVIG to SCIG, 77.1% commented on their health after the switch, with 33.3% reporting their health as the same and 62.9% reporting their health as better after the switch (Fig 3B).

## Discussion

We report a significant improvement in clinical outcomes including reduction in annual infections, ER visits, and hospitalizations in patients with SAD on IgRT. This study also reveals that a significant percentage of the patients with SAD on IgRT reported their health as better compared to before they started any IgRT. Nevertheless, despite very suggestive evidence, in the absence of specific SAD guidelines and recommendations, the initiation of IgRT remains a complex decision, which in part may reflect the heterogeneity of this patient population. While the current literature suggests that IgRT should be considered in patients with IgG levels <4 g/L and/or with history of serious and recurrent infections, there is a lack of consensus on the optimum target serum Ig levels in patients with SAD [4]. Similarly, a lack of clear definition of what constitutes severe and frequent infection in patients with SAD, who often have multiple advanced comorbidities, may hinder consensus integral to evaluation and management of these patients.

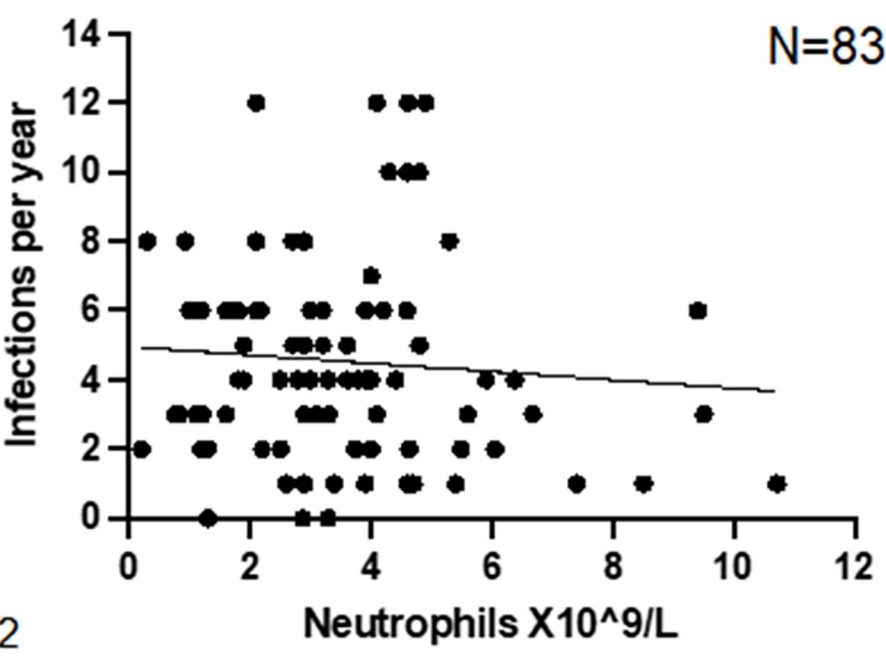

**Fig 2. Reported annual rate of infections vs neutrophils count at baseline.** The graph represents the number of infections per year reported by patients before starting IgRT, in relation to the measured neutrophils count. Spearman r = -0.075 [95% CI -0.292–0.1494].

Even when the decision to initiate IgRT is made, in the absence of SAD-specific recommendations, dosing and adjustments to the regimen are limited and thus mostly based on the PID literature. A starting dose of 0.4–0.8 g/kg every 4 weeks using actual body weight is an accepted practice, used in our centres as well [14, 15]. However, there has been some data suggesting

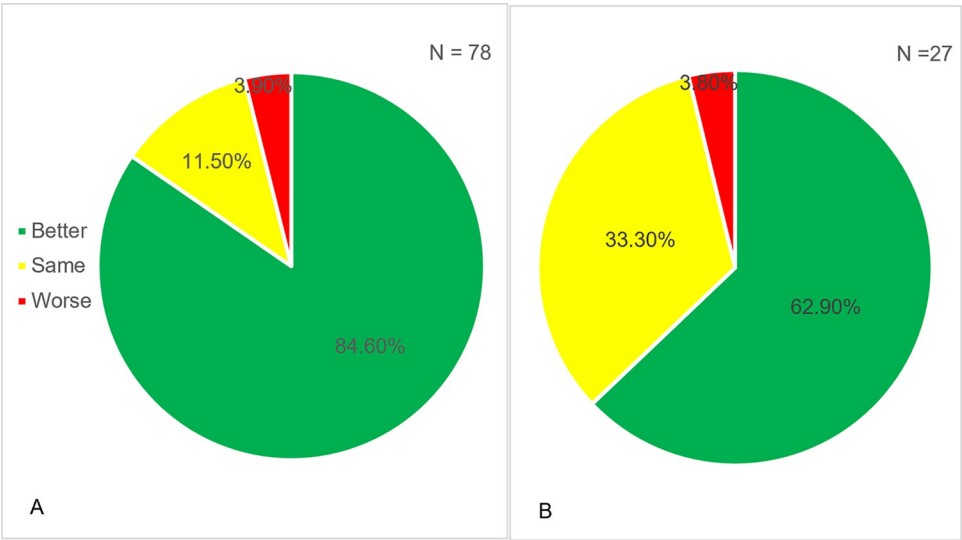

**Fig 3. Patient-reported health states.** Patients were asked how they perceived their overall health status: better (green), same (yellow) or worse (red). (A) in all patients before and after IgRT initiation. (B) in patients before and after switching from IVIG to SCIG.

that similar outcomes can be achieved with lower SCIG dosages and lower IgG trough levels in patients with SAD, perhaps in part due to the different underlying disease mechanisms and other patient factors contributing to antibody deficiencies in these patients [16]. Moreover, the antibody deficiency in some of the patients with SAD may be transient and reversible after the underlying primary disease is addressed. Therefore, once IgRT has been initiated, the optimal duration of therapy in these patients is another important consideration, which may not always be straightforward, such as with HSCT, where the recovery of immunity may not be easily determined [17].

Furthermore, while data for SCIG in PID patients suggests that it has advantages over IVIG, some of which are the stable trough levels of IgG seen with SCIG as well as the fewer systemic side effects, there are presently no specific guidelines or recommendations for SCIG use in SAD and only a very small number of published reports on the efficacy of SCIG in SAD patients [18]. Current literature suggests that IVIG and SCIG offer similar efficacy in reducing the rate of infections [16]. The data from this current study supports that SCIG is associated with improved general health state when compared with IVIG, with 33.3% of the patients reporting their health as the same and 62.9% of patients reporting their health as better compared to before the switch. Furthermore, patients who were initially started on SCIG reported an 82.5% decrease in infection rate and a 90.9% decrease in ER Visits and 80.0% decrease in hospitalizations as compared to pre-treatment.

These data are in keeping with a previous study of 33 patients with SAD which reported an overall improvement in quality-of-life using a modified version of SF-36 in patients switched from IVIG to SCIG therapy [19]. SCIG was associated with fewer systemic side-effects, more stable immunoglobulin trough levels and provided patients with the convenience of self-administration at home [16, 20, 21]. These issues should be raised with patients when discussing potential routes of IgRT. We hypothesize that SCIG in patients with SAD has the potential to improve patient compliance and considerably decrease costs for patients as the need to commute to hospitals monthly or take time off to get IVIG is avoided with home administration. As for the hospital and health-care sector, a cost-analysis study done in Canada found that the costs associated with SCIG were significantly lower than those with IVIG, attributed to fewer physician and hospital visits and shorter total nursing time required to infuse Ig [22]. Overall, the positive impact of SCIG on general health state reported by patients with SAD further supports that home-based SCIG programs should become more widely available and accessible.

In our study, there were 11 SAD patients who were SOT recipients. In the published literature, Ig deficiency developed in 45% of patients 1-year post-transplantation while there was severe Ig deficiency (IgG <4 g/L) in 15% of all transplant recipients [23]. The latter was associated with an increased risk of infection and 1-year all-cause mortality. A predictor of post-transplant severe hypogammaglobulinemia in lung transplant recipients is low pre-transplant Ig levels [24]. Frequency of hypogammaglobulinemia can be different in different types of SOT. For instance, SAD is prevalent in 63% of lung transplant recipients in the first-year post-transplant, compared to 45% and 30% of patients, at 3 and 12 months post-renal transplantation, respectively [23]. However, there were more kidney transplant than lung transplant recipients of IgRT in our cohort. Most of our cases reported here were from the Ottawa Hospital where kidney, but not lung transplantation is performed. The prevalence of hypogammaglobulinemia and infections decreased overtime post-transplant, but we observed that IgRT was started at 3.0 [2.0, 4.8] years after transplant suggesting that SAD was not recognized, and diagnostic delay. [25]. It is important to note that IgRT initiation is linked to the recurrence and severity of infections in this patient population rather than solely to measured Ig levels. Additional research should be done to identify predictive factors of SAD post-transplant and to study the efficacy of IgRT in SOT patients.

This study has some limitations. First, the average number of infections, hospitalizations and ER visits are subject to recall bias, as patients were being asked to give an estimated number of these events months or years after. Second, the accuracy in determining the diagnosis of the infections recalled by patients when no data was found in electronic medical records was also subject to reporting bias. Third, there was some missing information (e.g., medications used prior to IgRT, the year IgRT was started, Ig serum levels) particularly when IgRT was initiated long before the study enrolment. Fourth, the ONIT program is mandated to facilitate and monitor self-administered SCIG, thus, IVIG-treated SAD patients who were not known to ONIT program were not captured.

As the largest study of patients with SAD on IgRT in Canada, our data demonstrates that IgRT significantly improves clinical outcomes and the general health state of patients. In patients with SAD, SCIG should be discussed as another option to IVIG, as our data further suggests willingness to use SCIG in patients with SAD. This emphasizes the need for increased awareness of SCIG among health care providers, greater availability of SCIG and patient education on this mode of Ig delivery. Establishing specific and standardised treatment recommendations and guidelines on the use of IgRT in patients with SAD should be prioritized, for clinicians who treat conditions more often associated with SAD. These will include hemato-oncologists, rheumatologists, immunologists to name a few. With the growth of new and more effective immunomodulatory therapies, the prevalence of SAD is likely to continue to increase, emphasizing the need for further studies in this patient population with unmet care needs, and promoting early recognition of SAD among clinicians using these new immunomodulatory agents.

## Supporting information

**S1 File.**
(XLSX)

## Acknowledgments

The Ottawa Method Center for creating and maintaining the web based ONIT case registry.

## Author Contributions

**Conceptualization:** Juthaporn Cowan.

**Data curation:** Armin Abadeh, Sarah Shehadeh.

**Formal analysis:** Armin Abadeh, Sarah Shehadeh, Juthaporn Cowan.

**Methodology:** Juthaporn Cowan.

**Project administration:** Juthaporn Cowan.

**Supervision:** Juthaporn Cowan.

**Validation:** Stephen Betschel, Susan Waserman, Donald William Cameron.

**Visualization:** Sarah Shehadeh.

**Writing – original draft:** Armin Abadeh.

**Writing – review & editing:** Armin Abadeh, Sarah Shehadeh, Stephen Betschel, Susan Waserman, Donald William Cameron, Juthaporn Cowan.

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
