## [Decision Letter · Decision Letter 0]

8 Sep 2023

PONE-D-23-25320

Clinical Outcomes of Immunoglobulin Treatment for Patients with Secondary Antibody Deficiency: Data from the Ontario Immunoglobulin Treatment Case Registry

PLOS ONE

Dear Dr. Cowan

Thank you for submitting your manuscript to PLOS ONE. After careful consideration, we feel that it has merit but does not fully meet PLOS ONE’s publication criteria as it currently stands. Therefore, we invite you to submit a revised version of the manuscript that addresses the points raised during the review process.

We look forward to receiving your revised manuscript.

Kind regards,

Mehmet Baysal

Academic Editor

PLOS ONE

Journal Requirements:

2. Thank you for providing the following Funding Statement:  

I have read the journal's policy and the authors of this manuscript have the following competing interests: JC received honoraria and consultation fees from AstraZeneca, Merck, GSK, Sanofi, Genzyme, Takeda, CSL Behring, Octapharma, and Biogen. SB received advisory, speaker and committee fees or research funding from Astria, Canadian Blood Services, CSL Behring, Grifols, Ionis Pharmaceuticals, Kalvista Novartis, Octapharma, Pharvaris, Sanofi, and Takeda. SW received research funding, speaker and consultation fees from CAAIF, ALK Abello, Pfizer, Aimmune Schroeder Foundation, Sean Delaney foundation, GSK, Novartis, CSL Behring, Sanofi, Astrazaneca, Takeda, Teva, Medexus, Mylan, AbbVie, Miravohealth, Bausch Lomb, Avir Pharma and Covis. DWC received consultation and speaker fees from Takeda and CSL Behring. SS and AA have declared that no competing interests exist. 

We note that one or more of the authors is affiliated with the funding organization, indicating the funder may have had some role in the design, data collection, analysis or preparation of your manuscript for publication; in other words, the funder played an indirect role through the participation of the co-authors. 

If the funding organization did not play a role in the study design, data collection and analysis, decision to publish, or preparation of the manuscript and only provided financial support in the form of authors' salaries and/or research materials, please review your statements relating to the author contributions, and ensure you have specifically and accurately indicated the role(s) that these authors had in your study in the Author Contributions section of the online submission form. Please make any necessary amendments directly within this section of the online submission form.  Please also update your Funding Statement to include the following statement: “The funder provided support in the form of salaries for authors [insert relevant initials], but did not have any additional role in the study design, data collection and analysis, decision to publish, or preparation of the manuscript. The specific roles of these authors are articulated in the ‘author contributions’ section.” 

If the funding organization did have an additional role, please state and explain that role within your Funding Statement. 

Please also provide an updated Competing Interests Statement declaring this commercial affiliation along with any other relevant declarations relating to employment, consultancy, patents, products in development, or marketed products, etc.  

Reviewers' comments:

Reviewer's Responses to Questions

**Comments to the Author**

1. Is the manuscript technically sound, and do the data support the conclusions?

Reviewer #1: Yes

Reviewer #2: Yes

2. Has the statistical analysis been performed appropriately and rigorously? 

Reviewer #1: Yes

Reviewer #2: Yes

3. Have the authors made all data underlying the findings in their manuscript fully available?

Reviewer #1: Yes

Reviewer #2: No

4. Is the manuscript presented in an intelligible fashion and written in standard English?

Reviewer #1: Yes

Reviewer #2: Yes

5. Review Comments to the Author

Reviewer #1: I write regarding PONE-D-23-25320 entitled "Clinical Outcomes of Immunoglobulin Treatment for Patients with Secondary Antibody Deficiency: Data from the Ontario Immunoglobulin Treatment Case Registry ".

An observational study evaluating immunoglobulin replacement therapy in patients with secondary antibody deficiency. A total of 141 patients were evaluated and especially their quality of life and infection rates were evaluated. Indeed, the use of IgRT in clinical practice is controversial in terms of cost effectiveness and indications. Therefore, I think that this article meets the unmet need on an important issue. I have a few suggestions.

1. Firstly why were patients recruited in only 2 years (June 2020 -September 2022)?

2. In total, there were 141 patients in the study. While there were 9 patients using IVIG and 131 using SCIG, the other 1 patient ?

3. It would be more appropriate to specify the references after the point rather than before.

4. Which questionnaire was used as quality of life? should be specified by reference. Was the questionnaire used validated in your country?

5. The long version of SCIG should be written on line 83.

6. In Table-1, n should also be specified in the IgRT section.

7. Who should make the statistics of the article? If not, you should definitely seek help from a professional statistician. It was observed that parametric values were averaged.

Reviewer #2: Secondary immune disorders are far more prevalent and can be caused by various diseases and their treatment, certain medications and sometimes due to surgical procedures. Secondary antibody deficiencies are generally poorly defined and there are no guidelines for managing patients with this condition. In this respect, the study evaluating a large number of patients is important. In addition, the importance of national and international databases, where demographic, clinical data, treatment options and outcomes of patients with rare diseases are recorded, should be emphasized again.

There are some issues that require minor revision.

1. The number of patients is 141, 131 on SCIg, 9 on IVIg. Which route for Ig replacement is given to the remaining 1 patient?

2. The mean age of the study group is 66 years. What is the range of ages?

3. Vaccination responses to specific vaccines (the pneumococcal polysaccharide vaccine, anti-Hib,..) should be checked to give an indication regarding immune function, even if the patient, especially with hematologic malignancy, has not suffered from infections, as they may be at risk of severe sepsis. Did the authors check the vaccine responses before the initiation of IgG therapy?

4. What were the indications for IgRT in patients with normal IgG levels?

5. Did any of the patients experience severe bacterial infections (sepsis, osteomyelitis, meningitides,..) before and after Ig treatment?

6. Did any of the patients have accompanying severe or chronic complications (bronchiectasis, malnutrition, renal impairment,..)?

7. Were there any patients with protein loss (renal or intestinal)?

8. Did the authors make a statistical analysis to correlate the infection rates and neutropenia prior to IgRT?

9. The authors should emphasize the formulations of the chosen SCIg (10%, 16%, 20%, or facilitated 10%).

6. PLOS authors have the option to publish the peer review history of their article (what does this mean?). If published, this will include your full peer review and any attached files.

Reviewer #1: No

Reviewer #2: No

---

## [Author Response · Author response to Decision Letter 0]

20 Oct 2023

Comments by the academic editor:

1. When submitting your revision, we need you to address these additional requirements. Please ensure that your manuscript meets PLOS ONE's style requirements, including those for file naming. The PLOS ONE style templates can be found at https://journals.plos.org/plosone/s/file?id=wjVg/PLOSOne_formatting_sample_main_body.pdf and https://journals.plos.org/plosone/s/file?id=ba62/PLOSOne_formatting_sample_title_authors_affiliations.pdf. 

Response: Thank you. We have made changes accordingly.

2. Thank you for providing the following Funding Statement: 

I have read the journal's policy and the authors of this manuscript have the following competing interests: JC received honoraria and consultation fees from AstraZeneca, Merck, GSK, Sanofi, Genzyme, Takeda, CSL Behring, Octapharma, and Biogen. SB received advisory, speaker and committee fees or research funding from Astria, Canadian Blood Services, CSL Behring, Grifols, Ionis Pharmaceuticals, Kalvista Novartis, Octapharma, Pharvaris, Sanofi, and Takeda. SW received research funding, speaker and consultation fees from CAAIF, ALK Abello, Pfizer, Aimmune Schroeder Foundation, Sean Delaney foundation, GSK, Novartis, CSL Behring, Sanofi, Astrazaneca, Takeda, Teva, Medexus, Mylan, AbbVie, Miravohealth, Bausch Lomb, Avir Pharma and Covis. DWC received consultation and speaker fees from Takeda and CSL Behring. SS and AA have declared that no competing interests exist. 

We note that one or more of the authors is affiliated with the funding organization, indicating the funder may have had some role in the design, data collection, analysis or preparation of your manuscript for publication; in other words, the funder played an indirect role through the participation of the co-authors. If the funding organization did not play a role in the study design, data collection and analysis, decision to publish, or preparation of the manuscript and only provided financial support in the form of authors' salaries and/or research materials, please review your statements relating to the author contributions, and ensure you have specifically and accurately indicated the role(s) that these authors had in your study in the Author Contributions section of the online submission form. Please make any necessary amendments directly within this section of the online submission form. Please also update your Funding Statement to include the following statement: “The funder provided support in the form of salaries for authors [insert relevant initials], but did not have any additional role in the study design, data collection and analysis, decision to publish, or preparation of the manuscript. The specific roles of these authors are articulated in the ‘author contributions’ section.” 

If the funding organization did have an additional role, please state and explain that role within your Funding Statement. 

Please also provide an updated Competing Interests Statement declaring this commercial affiliation along with any other relevant declarations relating to employment, consultancy, patents, products in development, or marketed products, etc. 

Within your Competing Interests Statement, please confirm that this commercial affiliation does not alter your adherence to all PLOS ONE policies on sharing data and materials by including the following statement: "This does not alter our adherence to PLOS ONE policies on sharing data and materials.” (as detailed online in our guide for authors http://journals.plos.org/plosone/s/competing-interests ). If this adherence statement is not accurate and there are restrictions on sharing of data and/or materials, please state these. Please note that we cannot proceed with consideration of your article until this information has been declared.

Response: Thank you, we have made the adjustment to our financial disclosure and reported it on the cover letter. 

Response: Yes, all data will be provided upon acceptance.

Response: Thank you, this has been revised. No articles used in reference in our article were retracted.

Reviewer’s comments:

Reviewer #1: I write regarding PONE-D-23-25320 entitled "Clinical Outcomes of Immunoglobulin Treatment for Patients with Secondary Antibody Deficiency: Data from the Ontario Immunoglobulin Treatment Case Registry ". An observational study evaluating immunoglobulin replacement therapy in patients with secondary antibody deficiency. A total of 141 patients were evaluated and especially their quality of life and infection rates were evaluated. Indeed, the use of IgRT in clinical practice is controversial in terms of cost effectiveness and indications. Therefore, I think that this article meets the unmet need on an important issue. I have a few suggestions.

1. Firstly why were patients recruited in only 2 years (June 2020 -September 2022)? 

Response: Patient recruitment for the case registry is still ongoing as part of our program (Ontario Immunoglobulin Treatment program). However, to analyze data for this manuscript, Sep 30, 2022, was indicated as a cut-off date for data extraction. 

2. In total, there were 141 patients in the study. While there were 9 patients using IVIG and 131 using SCIG, the other 1 patient? 

Response: Our sincere apology of this error. The one patient was initially included because of secondary antibody deficiency criterion; however, we realized that the patient was never started on immunoglobulin replacement therapy by the time of data extraction. Therefore, we decided to remove the patient. The total number of patients is now 140, with 131 on SCIG and 9 on IVIG.

3. It would be more appropriate to specify the references after the point rather than before. 

Response: There were a few references written within the sentence, these were corrected. 

4. Which questionnaire was used as quality of life? should be specified by reference. Was the questionnaire used validated in your country? 

Response: The data included in this manuscript does not contain quality of life data. We changed the terminology of “quality of life” to “general health” where we asked if the patient felt that his/her overall health was the same, worse, or better at the time of study visit as compared to before IgRT. 

5. The long version of SCIG should be written on line 83. 

Response: Thank you. It has been corrected.

6. In Table-1, n should also be specified in the IgRT section. 

Response: Thank you. We have specified n in Table-1 as suggested. 

7. Who should make the statistics of the article? If not, you should seek help from a professional statistician. It was observed that parametric values were averaged. 

Response: We consulted a statistician. Median and interquartile range is reported for data that did not have a normal distribution. 

Reviewer #2: Secondary immune disorders are far more prevalent and can be caused by various diseases and their treatment, certain medications and sometimes due to surgical procedures. Secondary antibody deficiencies are generally poorly defined and there are no guidelines for managing patients with this condition. In this respect, the study evaluating a large number of patients is important. In addition, the importance of national and international databases, where demographic, clinical data, treatment options and outcomes of patients with rare diseases are recorded, should be emphasized again.

There are some issues that require minor revision.

1. The number of patients is 141, 131 on SCIg, 9 on IVIg. Which route for Ig replacement is given to the remaining 1 patient? 

Response: Our sincere apology of this error. The one patient was initially included because of secondary antibody deficiency criterion; however, we realized that the patient was never started on immunoglobulin replacement therapy by the time of data extraction. Therefore, we decided to remove the patient. The total number of patients is now 140, with 131 on SCIG and 9 on IVIG.

2. The mean age of the study group is 66 years. What is the age range? 

Response: Age range is between 23- 85 years old. 

3. Vaccination responses to specific vaccines (the pneumococcal polysaccharide vaccine, anti-Hib,..) should be checked to give an indication regarding immune function, even if the patient, especially with hematologic malignancy, has not suffered from infections, as they may be at risk of severe sepsis. Did the authors check the vaccine responses before the initiation of IgG therapy? 

Response: We followed the guidelines by starting secondary antibody deficient patients on IgRT when IgG are low, especially <4g/L or when there is a history of recurrent sinopulmonary infections (European Medicines Agency. Guideline on core SmPC for human normal immunoglobulin. Committee for Medicinal Products for Human Use). The results of the pneumococcal and hemophilus serology tests are not widely available in Ontario, unfortunately. The current turnaround time for the pneumococcal serology test result is more than one year in our region. This is not useful in clinical setting. 

4. What were the indications for IgRT in patients with normal IgG levels? 

Response: Normal IgG levels at our sites are between 7.0 and 16.0 g/L. Only two patients had normal IgG levels prior to initiation of therapy. The indication for treatment of the first patient was low IgG2 and low IgG3 levels. The patient was on chronic immunosuppressive therapy for severe scleroderma affecting both lungs and having undergone double lung transplant within a few years after starting IgRT. The other patient had an abnormal protein electrophoresis showing abnormal band in IgG kappa, with total gamma globulins of 5.0 despite total IgG showing a level of 8.6 g/L before IgRT. 

5. Did any of the patients experience severe bacterial infections (sepsis, osteomyelitis, meningitides,..) before and after Ig treatment?

Response: This is a very good point. There was a limitation in data collection for 131 patients who were already on IgRT prior to ONIT case registry enrollment. The data collected did not specify type of infections. We only recorded the overall number of infections, treatments for infections, ED visits, and hospitalizations based on patient’s report which was subjected to a recall bias. However, once the patients were in our program, we prospectively recorded types and severity of infections. We did not have any reported sepsis, osteomyelitis, or meningitides after starting on IgRT.

6. Did any of the patients have accompanying severe or chronic complications (bronchiectasis, malnutrition, renal impairment…)??

Response: Thank you. Two patients had concurrent bronchiectasis; ten patients had renal impairment. This was added to the results section under demographics. 

7. Were there any patients with protein loss (renal or intestinal)? 

Response: This is a good point. None of the 140 patients reported in this manuscript had evidence of renal or intestinal protein loss. In our patient population, the cause of secondary antibody deficiency is attributed to their primary hematologic malignancy or immune dysregulation and the immunosuppressive therapy associated with it. 

8. Did the authors make a statistical analysis to correlate the infection rates and neutropenia prior to IgRT?

Response: This is also a good point. Of 140 patients, only 83 had neutrophil counts prior to IgRT recorded. We graphed a scatter plot and analyzed the correlation between reported infection rate and neutrophil counts before IgRT. We found no correlation with spearman r = -0.075 [95% CI -0.292 – 0.1494]. This data should be confirmed prospectively in IgRT naïve patients who are newly enrolled in our ONIT registry going forward. 

9. The authors should emphasize the formulations of the chosen SCIg (10%, 16%, 20%, or facilitated 10%). 

Response: This was added to the results section under IgRT dosage, formulations, and treatment duration.

---

## [Decision Letter · Decision Letter 1]

31 Oct 2023

Clinical Outcomes of Immunoglobulin Treatment for Patients with Secondary Antibody Deficiency: Data from the Ontario Immunoglobulin Treatment Case Registry

PONE-D-23-25320R1

Dear Dr. Cowan

We’re pleased to inform you that your manuscript has been judged scientifically suitable for publication and will be formally accepted for publication once it meets all outstanding technical requirements.

Kind regards,

Mehmet Baysal

Academic Editor

PLOS ONE

Additional Editor Comments (optional):

Reviewers' comments:

Reviewer's Responses to Questions

**Comments to the Author**

1. If the authors have adequately addressed your comments raised in a previous round of review and you feel that this manuscript is now acceptable for publication, you may indicate that here to bypass the “Comments to the Author” section, enter your conflict of interest statement in the “Confidential to Editor” section, and submit your "Accept" recommendation.

Reviewer #1: All comments have been addressed

Reviewer #2: (No Response)

2. Is the manuscript technically sound, and do the data support the conclusions?

Reviewer #1: Partly

Reviewer #2: Yes

3. Has the statistical analysis been performed appropriately and rigorously? 

Reviewer #1: Yes

Reviewer #2: Yes

4. Have the authors made all data underlying the findings in their manuscript fully available?

Reviewer #1: Yes

Reviewer #2: Yes

5. Is the manuscript presented in an intelligible fashion and written in standard English?

Reviewer #1: Yes

Reviewer #2: Yes

6. Review Comments to the Author

Reviewer #1: The authors accepted my suggestions and made the necessary edits. It is acceptable in its current state.

Reviewer #2: (No Response)

7. PLOS authors have the option to publish the peer review history of their article (what does this mean?). If published, this will include your full peer review and any attached files.

Reviewer #1: No

Reviewer #2: No

---

## [Editor Report · Acceptance letter]

7 Nov 2023

PONE-D-23-25320R1 

Clinical outcomes of immunoglobulin treatment for patients with secondary antibody deficiency: Data from the Ontario immunoglobulin treatment case registry 

Dear Dr. Cowan:

I'm pleased to inform you that your manuscript has been deemed suitable for publication in PLOS ONE. Congratulations! Your manuscript is now with our production department. 

Kind regards, 

on behalf of

Dr. Mehmet Baysal 

Academic Editor

PLOS ONE